# Land Equivalent Ratio in the Intercropping of Cucumber with Lettuce as a Function of Cucumber Population Density

**Rodolfo Gustavo Teixeira Ribas [1], Arthur Bernardes Cecílio Filho [2,*],**
**Alexson Filgueiras Dutra [2], José Carlos Barbosa [2] and Glauco de Souza Rolim [2]**

1    Federal Institute of Education, Science and Technology of Rondônia (IFRO), Cacoal 76960-970, Brazil;
     rodolfo.ribas@ifro.edu.br

2    Agricultural Production Sciences Department, São Paulo State University (UNESP), Jaboticabal 14.884-900,
     Brazil; alexsondutra@gmail.com (A.F.D.); jc.barbosa@unesp.br (J.C.B.); glauco.rolim@unesp.br (G.d.S.R.)

\*    Correspondence: arthur.cecilio@unesp.br; Tel.: +55-16-3209-7377

**Abstract:** Lettuce and cucumber are two important vegetables cultivated in greenhouses. Intercropping can increase the yield without increasing the demands for inputs. A more efficient use of resources in production systems can reduce costs and environmental impacts. We evaluated the land equivalent ratio (LER) of intercropping cucumber and lettuce as a function of the cucumber population. An experiment was conducted in a greenhouse to evaluate the cucumber population density (100, 85, 70, and 55% of 2.35 plants $m^{-2}$) and two lettuce cultivars, 'Lucy Brown' and 'Vanda'. The cucumber population density affected the amount of photosynthetically active radiation that reached the lettuce. The higher the density, the lower the total fresh mass and yield of the two lettuce cultivars. Fruit yield per plant and per area decreased and increased, respectively, as the density increased. LER was highest when cucumber was intercropped with 'Vanda' lettuce. LER increased with the density of 'Vanda' but decreased for 'Lucy Brown'. 'Lucy Brown' produced commercial traits (head formation) only at the lowest density (55%). The presence of lettuce did not affect the cucumber yield per plant or per area. The intercropped system used land more efficiently than monocultured crops of lettuce and cucumber, with better results for 'Vanda' than 'Lucy Brown'.

**Keywords:** *Lactuca sativa* L.; *Cucumis sativus* L.; photosynthetically active radiation; production system

## 1. Introduction

Intercropping is a technology that enables production with rational land use and less environmental impact [1]. Besides, intercropping presents higher biological diversity and rapid soil cover [2], allows greater efficiency in the use of agricultural inputs and labor, and helps increase the income from agricultural activity [3]. However, intercropping is a more complex production system than monoculture [1], and its efficiency depends directly on the species and management practices [4,5]. Lettuce and cucumber differ in cycle length, size, architecture [3], light demand [6,7], nutrient demand, and other important traits in intercropping systems. These differences allow optimizing the complementarity between the species, both temporally and spatially, with a better use of available resources, thereby minimizing possible interspecific competition [8].

Rezende et al. [9,10] and Cecílio Filho et al. [3] reported that the feasibility of intercropping cucumber with lettuce depended on the time of year, time of lettuce transplantation (season of establishment of the intercropping), and the cultivar of lettuce. Intercropping these two species is impractical in a hot and rainy climate because it favors the rapid growth of cucumber, which

substantially shades the lettuce and thereby interferes with its growth. The prices of these vegetables are nonetheless highest during this period, so we reevaluated the interaction between the species as a function of the cucumber population. Lower cucumber planting densities should allow more solar radiation to reach the lettuce and consequently increase spatial complementarity and land equivalent ratio (LER) due to the better performance of the lettuce compared with the fixed population of 2.35 plants m$^{-2}$ used by Rezende et al. [9,10] and Cecílio Filho et al. [3].

The population of the main crop in additive intercropping was kept constant, and individuals of the other crop were added to that population. We added the cucumber crop to the main crop (lettuce) with a system of vertical, guided growth. Reducing interspecific competition for environmental resources, unlike monocultures with only intraspecific competition, is the largest challenge for the success of intercropping these two vegetables of great economic importance worldwide. We thus targeted spatial and/or temporal complementarity. Heuvelink [11] and Mattera et al. [12] reported that population density affected canopy structure, light interception, radiation-use efficiency, and consequently biomass production of tomato and lucerne.

According to the above, it is possible to present the following hypotheses: (i) the lettuce will not affect the cucumber yield regardless of the cucumber spacing; (ii) the bigger the cucumber spacing, the higher the solar radiation to reach lettuce; (iii) the higher the lettuce yield, the higher the efficiency of land use.

This study thus evaluated land equivalent ratio in the intercropping of lettuce with cucumber as a function of cucumber planting density.

## 2. Materials and Methods

### 2.1. Experimental Area

The experiment was carried out in a greenhouse at São Paulo State University (21°14'39" S, 48°17'10" W; 575 meters above sea level) from 30 January to 5 April 2015. A meteorological mini-station was installed in the center of the greenhouse at a height of 2.0 m to record the temperature, relative air humidity and photosynthetically active radiation (PAR) every 5 minutes. The mean temperature was 26 °C (range: 16.1–37.2 °C), with mean maximum and minimum temperatures of 32.5 and 19.6 °C, respectively. The mean relative air humidity was 74.5% (range: 30–100%), with mean maximum and minimum humidities of 98 and 50%, respectively. PAR during the experimental period is shown in Figure 1.

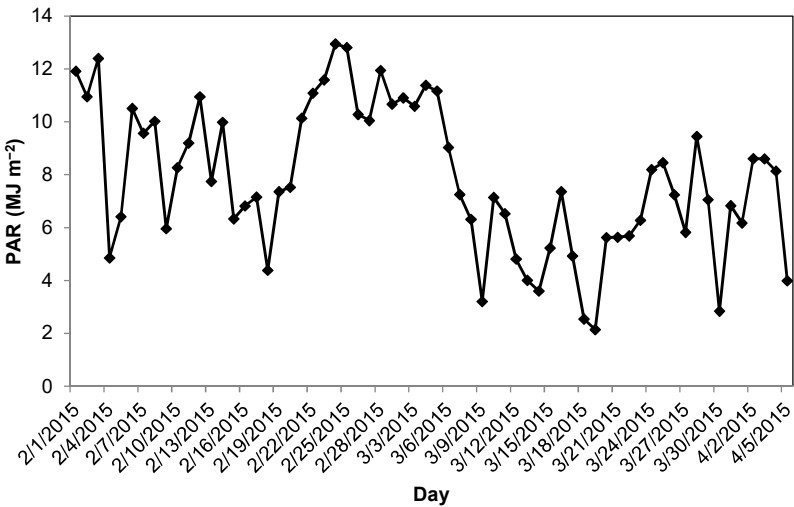

**Figure 1.** Daily photosynthetic active radiation registered in the greenhouse.

## 2.2. Treatments and Experimental Design

Fourteen treatments were evaluated in a randomized complete block design, with four replicates in a $4 \times 2 + 6$ factorial scheme. Eight of the treatments were combinations of two factors: four cucumber population densities (100, 85, 70, and 55% of 2.35 plants $m^{-2}$) and two lettuce cultivars ('Lucy Brown', Crisphead group, and 'Vanda', Loose leaf group). The other six treatments were four cucumber monocultures of the populations, for identifying a possible effect of planting density, and two monocultures of the lettuce cultivars (Table 1). Lettuce was the main crop, and cucumber was the secondary crop.

**Table 1.** Characterization of monoculture treatments and intercrop of two lettuce cultivars and four cucumber population densities.

| Treatment | Acronyms | Lettuce 'Lucy Brown' | Lettuce 'Vanda' | Cucumber [1] |
|---|---|---|---|---|
| 1—Intercropping | LBI100 | Presence | Absent | Presence (100%) |
| 2—Intercropping | LBI85 | Presence | Absent | Presence (85%) |
| 3—Intercropping | LBI70 | Presence | Absent | Presence (70%) |
| 4—Intercropping | LBI55 | Presence | Absent | Presence (55%) |
| 5—Monoculture | SM100 | Absent | Absent | Presence (100%) |
| 6—Monoculture | SM85 | Absent | Absent | Presence (85%) |
| 7—Monoculture | SM70 | Absent | Absent | Presence (70%) |
| 8—Monoculture | SM55 | Absent | Absent | Presence (55%) |
| 9—Monoculture | LBM | Presence | Absent | Absent |
| 10—Intercropping | VI100 | Absent | Presence | Presence (100%) |
| 11—Intercropping | VI85 | Absent | Presence | Presence (85%) |
| 12—Intercropping | VI70 | Absent | Presence | Presence (70%) |
| 13—Intercropping | VI55 | Absent | Presence | Presence (55%) |
| 14—Monoculture | VM | Absent | Presence | Absent |

[1] The value in parentheses refers to the percentage of the population of cucumber in relation to 2.35 plant $m^{-2}$.

The experimental units of the intercrops and monocultures of cucumber for 100, 85, 70, and 55% of the total population (2.35 plants $m^{-2}$) were 3.0, 3.6, 4.3, and 5.4 m in length, respectively. The units were 1.70 m wide, corresponding to the width of the plant bed and half the distance between adjacent beds. The traits of the lettuce and cucumber were evaluated for all plants except the first and last plants in each row.

## 2.3. Installation of the Experiment

The soil was limed and fertilized based on an initial analysis. The monocultured lettuce and cucumber crops were planted and cover fertilized as recommended by Trani et al. [13,14]. For the intercroppings we applied 40 kg $ha^{-1}$ N (urea), 400 kg $ha^{-1}$ $P_2O_5$ (simple superphosphate), and 200 kg $ha^{-1}$ $K_2O$ (potassium chloride) according to cucumber demand, which is higher than that of lettuce. At 20, 40, and 60 days after transplanting, a total of 150 kg $ha^{-1}$ N and 120 kg $ha^{-1}$ $K_2O$, in the same sources, were applied to cucumber crop, and 90 kg $ha^{-1}$ N to lettuce, which was splitted to 10, 20 and 30 days after transplanting.

Lettuce and cucumber seedlings were transplanted on the same day to the plant beds. The row $\times$ plant spacings were $0.35 \times 0.35$ m for 'Lucy Brown' lettuce and $0.25 \times 0.25$ m for 'Vanda'. The 'Soldier' cucumber cultivar was grafted onto 'Keeper' squash and transplanted into a plot with 1.20 m between double rows and 0.50 m between rows. The spacings between cucumber plants in a row were 0.50, 0.59, 0.72, and 0.91 m to obtain population densities of 100, 85, 70, and 55% of 2.35 $m^{-2}$ plants, respectively. The plants were guided vertically with plastic tape on a single stem, and axillary branches were cut to a height of 0.4 m. The shoots were then allowed to develop and were fixed to wires parallel to the ground and 0.40 m apart. The apical meristems of the lateral shoots were cut when the shoots had

two fruits and three leaves. The apical meristems of the main shoots were cut when the plants had 19 nodes. The plants were irrigated by tube drippers.

*2.4. Characteristics Evaluated Percent of Photosynthetic Assimilation Ratio (PAR)*

### 2.4.1. Percent PAR

PAR during lettuce and cucumber intercropping was evaluated at two heights: 2.0 and 0.15 m above the cucumber canopy (PAR A and PAR B, respectively). PAR B and A for each population were recorded daily at 12:00 at the centers of four cucumber plants and between two plants. The means of the two points at 0.15 m and the percentage corresponding to PAR A (above the cucumber plants) were calculated.

The equation with the best fit was obtained using the percentage of PAR B/A for each density as a function of time.

### 2.4.2. Lettuce Fresh Mass

Total lettuce fresh mass (g plant$^{-1}$): 'Vanda' and 'Lucy Brown' were harvested at 41 and 56 days after transplantation, respectively.

Commercial lettuce fresh mass (g plant$^{-1}$): the plants considered as commercial had specific traits of color and leaf shape and were not rotting or etiolated. For 'Vanda' lettuce, the entire shoot was considered as commercial mass, whereas only the head was included for 'Lucy Brown', excluding the leaves external to the head.

### 2.4.3. Lettuce and Cucumber Yield

Total lettuce yield (kg m$^{-2}$): estimated using the sum of the total fresh masses of the plants in the sampled area.

Commercial lettuce yield (kg m$^{-2}$): estimated using the sum of the fresh commercial masses of the plants in the sampled area.

Commercial cucumber yield (g plant$^{-1}$ and kg m$^{-2}$): commercial-class cucumbers (class 20: 20–25 cm in length) were harvested three times a week between 3 March and 4 April, for a total of 15 harvests. The commercial fruits had inclination angles <30° to the longitudinal axis of the fruit and were not physically damaged, as defined by the classification by HortiBrasil [15].

### 2.4.4. Number of Commercial Cucumbers

The number of commercial fruits (fruits m$^{-2}$) was determined by the sum of class-20 fruits harvested during the cycle.

### 2.4.5. Land Equivalent Ratio (LER)

LER was calculated using the equation proposed by Willey and Osiru [16]: LER = $(Y_{12}) / (Y_{11})$ + $(Y_{21}) / (Y_{22})$, where $Y_{12}$ is the yield of crop 1 intercropped with crop 2, $Y_{21}$ is the yield of crop 2 intercropped with crop 1, $Y_{11}$ is the yield of monocultured crop 1, and $Y_{22}$ is the yield of monocultured crop 2. Thus, the LER expresses how much land in a monoculture system is needed to produce the same amount of food in intercropping system.

### 2.4.6. Relative Yield (RY)

The index for the RYs of the monocultured and intercropped crops [17] was calculated as $RY_1 = Y_{12} / Y_{11}$ and $RY_2 = Y_{21} / Y_{22}$, where $Y_{12}$ is the yield of crop 1 intercropped with crop 2, and $Y_{21}$ is the yield of crop 2 intercropped with crop 1. The LER and RY indices were obtained using the total yields of 'Vanda' and 'Lucy Brown' and the commercial cucumber yield (kg m$^{-2}$) for each population density. The denominator (monoculture) corresponding to the average of monocultures (plots) was used for each population density, as proposed by Federer [18].

*2.5. Statistical Analyses*

The PAR data were verified using the significance of the equations for each curve. The regressions were nonlinear, so each curve was linearized by logX and logY, the parallelism was verified using a *t*-test, and the coincidence between lines was verified using an *F* test. For lettuce, the analysis of variance followed the randomized block design in a $2 \times 4 + 2$ factorial scheme. The two lettuce cultivars ('Vanda' and 'Lucy Brown'), four cucumber populations (100, 85, 70, and 55% of 2.35 m$^{-2}$ plants), and the monocultured 'Vanda' and 'Lucy Brown' lettuce were analyzed. For cucumber, the analysis of variance followed a randomized block design in a $3 \times 4$ factorial scheme. Three cultivation systems (cucumber intercropped with 'Vanda', cucumber intercropped with 'Lucy Brown', and monocultured cucumber) and four population densities (100, 85, 70, and 55% of 2.35 plants m$^{-2}$) were analyzed. An analysis of variance of the LER data was performed using a randomized block design in a $2 \times 4$ factorial scheme. The two lettuce cultivars ('Vanda' and 'Lucy Brown') and four intercropped cucumber population densities (100, 85, 70, and 55% of 2.35 plants m$^{-2}$) were analyzed. The total yield was used for calculating LER of the intercroppings with 'Vanda' and 'Lucy Brown' lettuce.

The data were submitted to an analysis of variance using the F test, and the means were then compared using Tukey's test at 5% probability. Significant effects of population density and the interaction between cropping system and population density were identified by polynomial regression analyses. The interaction between the additional treatments (monocultures) and the factorial, when significant, were identified and analyzed by contrasts; the monocultured lettuce was contrasted with the intercropped crops for each cultivar and density of the intercropped cucumber population. AgroEstat [19] was used for all analyses.

## 3. Results

*3.1. PAR*

The PAR available for the lettuce intercropped with cucumber was lower at 0.15 m than 2 m (above the cucumber plants), regardless of the cucumber population density (Figure 2). The linearization of the PAR equations indicated that the parallelism test did not differ significantly, but the line coincidence test indicated that the line for a density of 55% differed significantly from the line for a density of 100% (Table 2).

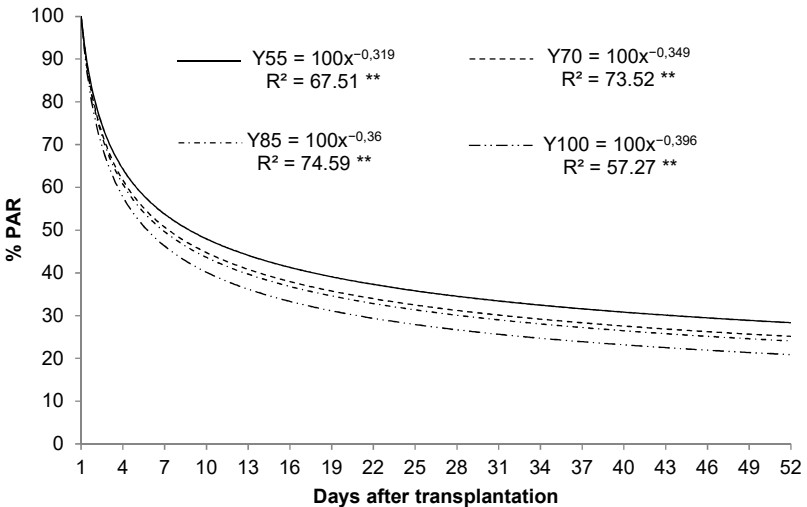

**Figure 2.** Percentage of photosynthetic active radiation (PAR) that reached the canopy of lettuces for the four population densities of the cucumber. ** Significant equations at 1% in the F test.

**Table 2.** *T*-test for parallelism and *F* test for coincidence between the straight lines, from the linearized equations for the photosynthetically active radiation.

| Population Density | T-test (Parallelism) | | |
|:---:|:---:|:---:|:---:|
| | Population Density | | |
| | **55** | **70** | **85** |
| 70 | 0.5683 [ns1] | – | – |
| 85 | 0.7710 [ns] | 0.2092 [ns] | – |
| 100 | 1.0795 [ns] | 0.6648 [ns] | 0.5096 [ns] |
| | F test (Coincidence) | | |
| | 55 | 70 | 85 |
| 70 | 1.5558 [ns2] | – | – |
| 85 | 2.6858 [ns] | 0.1638 [ns] | – |
| 100 | 7.0390 ** | 3.0562 [ns] | 2.1009 [ns] |

[1] T-test. ns—not significant; [2] F test. ns—not significant; ** $p \leq 0.01$.

### 3.2. Lettuce

The cucumber population density affected the total fresh mass of the monocultured ($p < 0.01$) and intercropped ($p < 0.01$) lettuce cultivars. The total fresh mass differed between the monocultured lettuce cultivars ($p < 0.01$). There was a significant interaction between additional treatments (monocultures) and cultivar and density (Table 3). The total fresh mass did not differ significantly between the lettuce cultivars 'Vanda' and 'Lucy Brown' when intercropped with cucumber but differed significantly between the monocultures. Monocultured total fresh mass was 91% higher for 'Lucy Brown' than 'Vanda' (Table 3). The higher the cucumber population density, the lower the total fresh mass of the two intercropped lettuce cultivars. 'Lucy Brown', however, was more sensitive to increased shading, due to the higher number of cucumber plants per unit area. Total fresh mass at a density of 74%, equivalent to 1.74 plants m$^{-2}$, was lower for 'Lucy Brown' than 'Vanda' (Figure 3). The total fresh masses of 'Vanda' and 'Lucy Brown' were adversely affected by intercropping, producing less mass than when grown in monoculture (Table 4).

**Table 3.** Variance analysis for total fresh mass (TFM) and total yield (TY) in lettuces 'Vanda' and 'Lucy Brown' as a function of lettuce (L) cultivars and cucumber population density (PD).

| Source of Variation | TFM | TY |
|:---:|:---:|:---:|
| L | 1.24 [ns2] | 74.92 ** |
| PD | 23.35 ** | 11.84 ** |
| L × PD | 5.92 ** | 0.40 [ns] |
| AT[1] | 136.34 ** | 0.17 [ns] |
| (L × PD) × AT | 399.00 ** | 25.52 ** |
| CV (%) | 14.46 | 16.14 |

| Lettuce | Intercropping | | Monoculture | |
|:---:|:---:|:---:|:---:|:---:|
| | TFM<br>g plant$^{-1}$ | TY<br>kg m$^{-2}$ | TFM<br>g plant$^{-1}$ | TY<br>kg m$^{-2}$ |
| 'Vanda' | 133.96 A [3] | 2.14 A | 237.29 B | 3.80 A |
| 'Lucy Brown' | 144.21 A | 1.12 B | 452.50 A | 3.70 A |

[1] AT = Additional treatment: 'Vanda' and 'Lucy Brown' monocultures; [2] F test. ns—not significant, ** $p \leq 0.01$; [3] Averages followed by the same letter in the column do not differ from each other in F test ($p > 0.05$).

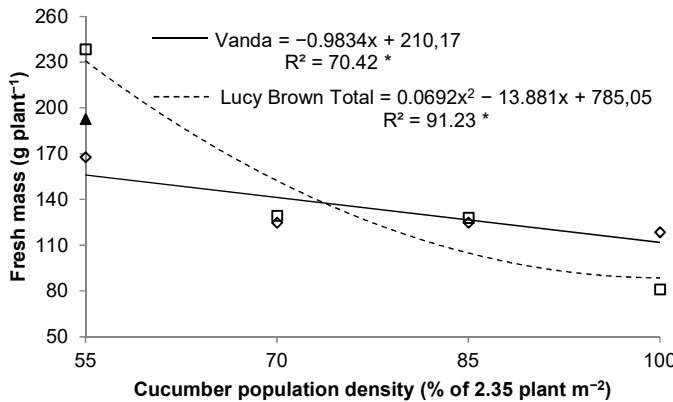

**Figure 3.** Total fresh mass (= commercial) of 'Vanda' and total fresh commercial mass of 'Lucy Brown' as a function of the population density of the cucumber. ◊ 'Vanda'; □ 'Lucy Brown' Total; ▲ 'Lucy Brown' Commercial. * $p \leq 0.05$ in F test.

**Table 4.** Groups of contrasts for total fresh mass (TFM) and total yield (TY) of 'Vanda' and 'Lucy Brown' lettuces between additional treatments ('Vanda' and 'Lucy Brown' monocultures) and factorial (2 lettuces × 4 densities of cucumber in the intercropping).

| Contrasts | TFM | TY |
|---|---|---|
| VM × VI100 [1] | 118.75 [2] ** | 1.90 [3] ** |
| VM × VI85 | 112.50 ** | 1.80 ** |
| VM × VI70 | 112.50 ** | 1.80 ** |
| VM × VI55 | 69.58 ** | 1.11 ** |
| LBM × LBI100 | 371.46 ** | 3.04 ** |
| LBM × LBI85 | 324.58 ** | 2.65 ** |
| LBM × LBI70 | 323.13 ** | 2.64 ** |
| LBM × LBI55 | 214.00 ** | 1.95 ** |

[1] Description of acronyms in Table 1; [2] Difference between the means of the treatments listed in the first column of the table; [3] ** significant at $p \leq 0.01$, according to F test.

All 'Vanda' plants had commercial quality, regardless of the cucumber population density, although the head size decreased as the density increased (Figure 3). The commercial fresh mass of monocultured 'Vanda' was 237.29 g. Commercial traits for 'Lucy Brown', however, were only obtained at the lowest cucumber population density, 55% of 2.35 plants m$^{-2}$, corresponding to 1.3 plants m$^{-2}$, with heads forming only at this density, for a commercial fresh mass of 192.71 g. The commercial fresh mass of monocultured 'Lucy Brown' was 405.21 g. Population density and lettuce cultivar affected the total yield ($p < 0.01$), and none of the factors interacted significantly (Table 4).

Total yield was 91% higher for intercropped 'Vanda' than 'Lucy Brown' lettuce, regardless of the cucumber population density (Figure 4 and Table 4), although the total fresh mass did not differ significantly between the cultivars. The difference in total yields was due to the lettuce spacing, which was smaller for 'Lucy Brown'. Total yield for both lettuces was affected by the higher cucumber population densities. Total yields were highest at the lowest density (55% of 2.35 plants m$-^2$), at 1.58 and 2.50 kg m$^{-2}$ for 'Lucy Brown' and 'Vanda', respectively (Figure 4). The total yield of monocultured lettuce did not differ significantly between 'Vanda' and 'Lucy Brown', which produced 3.8 and 3.7 kg m$^{-2}$, respectively (Table 4). Total yields were higher for the monocultured than the intercropped cultivars.

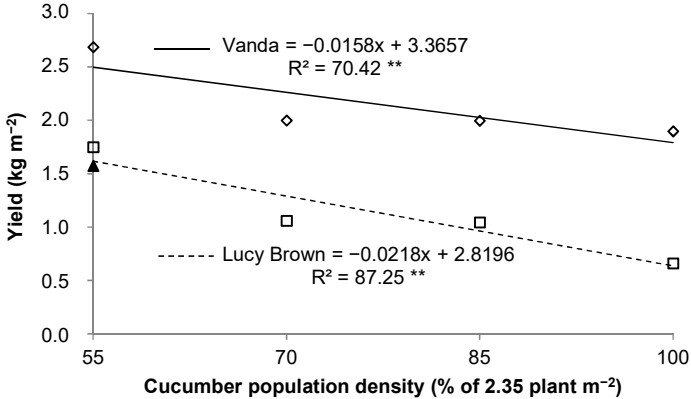

**Figure 4.** Total yield (= commercial) of 'Vanda' and total and commercial yield of 'Lucy Brown' as a function of cucumber population densities. ◊ 'Vanda'; □ 'Lucy Brown' Total; ▲ 'Lucy Brown' Commercial. ** $p \leq 0.01$ in F test.

The commercial yields of the cultivars were similar to the commercial fresh masses. The commercial yield of 'Lucy Brown' was highest at the lowest cucumber population density, corresponding to 1.57 kg m$^{-2}$. All 'Vanda' plants had commercial quality, so the commercial yield was equal to the total yield.

### 3.3. Cucumber

The cropping system did not affect ($p > 0.05$) the fruit yield per plant or per area, but the cucumber population density affected ($p < 0.01$) all traits (Table 5). The number of fruits per plant and per area decreased and increased, respectively, as the cucumber population density increased (Figure 5), and because the fruits were harvested at the same size, the number of fruits was the only component that affected the cucumber yield. Thus, the cucumber yield showed the same response to that observed for the number of fruits in Figure 6.

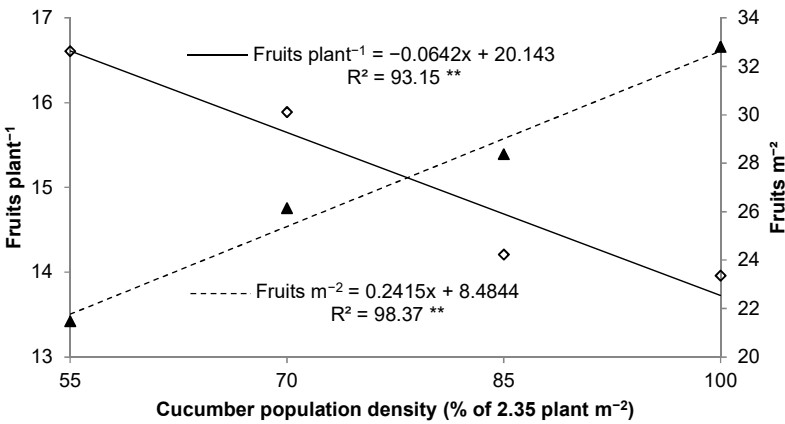

**Figure 5.** Number of fruits per plant and area of cucumber as a function of population densities. ◊ Fruits plant$^{-1}$; ▲ Fruits m$^{-2}$. ** $p \leq 0.01$ in F test.

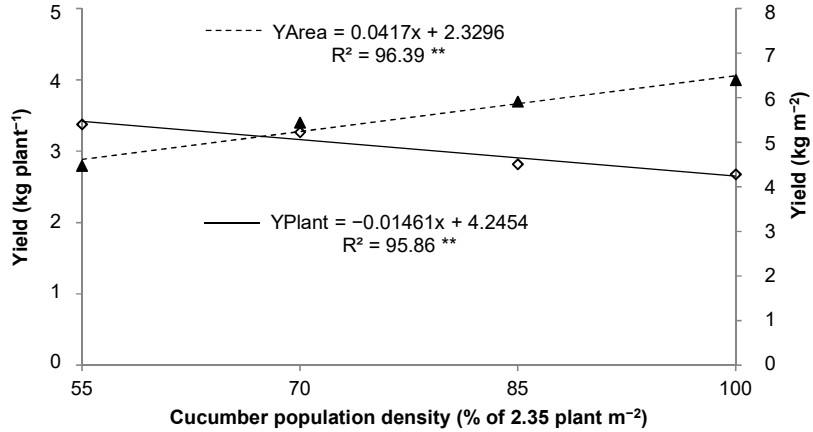

**Figure 6.** Cucumber yield by plant and by area as a function of population densities. ◊ kg plant$^{-1}$; ▲ kg m$^{-2}$. ** $p \leq 0.01$ in F test.

**Table 5.** Variance analysis for number of commercial fruits per plant (NFP) and area (NFA), commercial yield per plant (CYP) and area (CYA) of 'Soldier' cucumber, and land equivalent ratio (LER) and relative yield (RY$_{let}$ and RY$_{cuc}$) according to cultivation system (CS) and population density (PD).

| Source of Variation | NFP | NFA | CYP | CYA | LER | RY$_{let}$ | RY$_{cuc}$ |
|---|---|---|---|---|---|---|---|
| CS | 1.11 [ns1] | 1.17 [ns] | 0.78 [ns] | 0.82 [ns] | 30.82 **,[1] | 117.98 ** | 0.36 [ns] |
| PD | 6.42 ** | 25.78 ** | 6.57 ** | 14.10 ** | 1.18 [ns] | 25.25 ** | 10.59 ** |
| CS × PD | 1.46 [ns] | 1.39 [ns] | 1.22 [ns] | 1.48 [ns] | 3.51 * | 1.34 [ns] | 2.38 [ns] |
| CV (%) | 11.64 | 11.83 | 12.54 | 13.38 | 7.98 | 15.01 | 9.23 |

[1] F test. ns—not significant; ** $p \leq 0.01$; * $p \leq 0.05$.

### 3.4. LER and RY

LER was affected ($p < 0.05$) by the factors lettuce cultivars and cucumber population density. RY for lettuce (RY$_{let}$) was influenced by the cropping system and population density ($p < 0.01$) (Table 5). All LERs for intercropping were >1, indicating superiority over monoculturing. LER was highest when cucumber was intercropped with 'Vanda' lettuce. The LER indices were adjusted to the linear equation. The LER's for intercropping cucumber with 'Vanda' and 'Lucy Brown' linearly increased and linearly decreased, respectively, as the cucumber population density increased (Figure 7). RY for cucumber was influenced only by population density ($p < 0.01$) (Table 5). RY for both lettuces decreased, and RY for cucumber increased, with increasing cucumber population density (Figure 8).

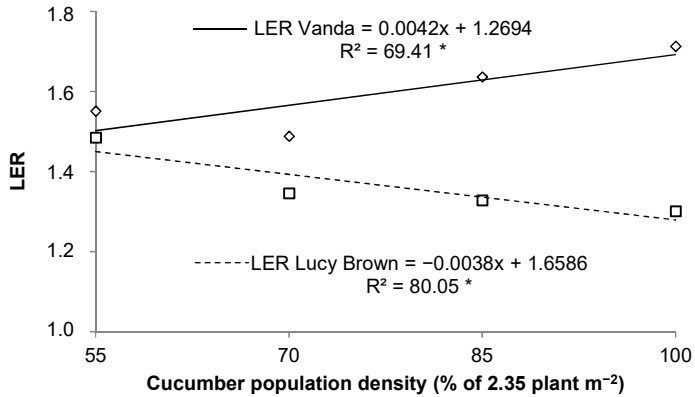

**Figure 7.** Land equivalent ratio (LER) of the intercropping of 'Vanda' and 'Lucy Brown' with cucumber. ◊ LER 'Vanda'; □ LER 'Lucy Brown'. * $p \leq 0.05$ in F test.

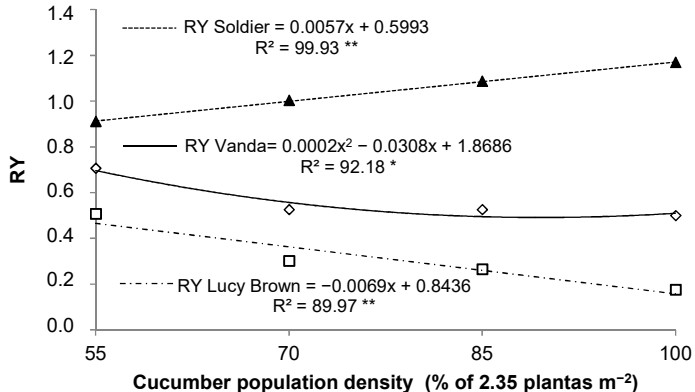

**Figure 8.** Relative yield (RY) of the intercropping components of 'Vanda' and 'Lucy Brown' with cucumber. ◇ RY 'Vanda'; □ RY 'Lucy Brown'; ▲ RY 'Soldier'. * $p \leq 0.05$; **: $p \leq 0.01$ in F test.

## 4. Discussion

Cucumber growth strongly negatively interfered with lettuce growth. The shading of the lettuce plants by the cucumber canopy reduced the passage of solar radiation. Only about 20–35% of the PAR reached the lettuce plants (Figure 2). The amounts of PAR that reached 'Vanda' and 'Lucy Brown' were higher when the cucumber population was low (55% of 2.35 plants m$^{-2}$), which directly influenced the lettuce production. PAR penetration for the lettuce plants and the fresh mass per plant and per area of both lettuce cultivars (Figure 3) decreased as the population density increased. The amount of PAR that arrived was not enough for the plants to satisfactorily produce biomass. Cucumber biomass was guided vertically, so its leaves were on a higher plane than the lettuce plants, causing shade and competition for light. This example indicates that less light decreases the yield [7,9,20,21].

The growth and yield of plants depend on the amount of PAR that reaches the plants [21–24]. Light is the main factor in the competition between plants in intercropping systems [25,26]. Differences in height, population density, and foliage inclination can all affect light interception and establish different levels of competition for light [26].

Nutrients and organic compounds may be translocated from one part of a plant to another under stress situations (nutritional, light, thermal, or water availability). Temperature and irrigation in our study did not differ between the intercropped and monocultured systems, with only light (PAR) being the limiting factor. Poorter et al. [27] reported that light had the largest impact on plant growth, especially when plants coexist. These authors observed that >30% of the variables responsible for growth decreased as the amount of light decreased, more than for other disturbances (water availability, thermal and nutritional disturbances).

Lettuce is sensitive to decreases in light [7,9,24,28,29]. The production of fresh mass and yield of the intercropped lettuces in our study were affected by the cucumber population density (Figures 3 and 4). Higher densities decreased the fresh masses and yields of the lettuce plants, corroborating the results obtained by Rezende et al. [9], and in the intercropping of cucumber with lettuce. Shading also had a negative effect when lettuce was intercropped with (guided) tomato relative to monocultured lettuce, with damage increasing with the delay in transplanting lettuce after tomato [30] and after cucumber [3].

The fruit yield per plant for cucumber was highest at the lowest planting density (Figure 5), which could be attributed to the larger amount of light (PAR) reaching the lower parts of the plants (Figure 2), where net photosynthesis is higher [31]. Cucumber is sensitive to shade [32], and more radiation provides a higher yield [32,33]. The fruit yield per area in our study, however, was higher at the highest density (2.35 plants m$^{-2}$), thereby increasing the yield per area [31,34].

Cucumber yields (per plant and per area) were not affected by the presence of the lettuce, as also reported by Cecílio Filho et al. [3] when evaluating cucumber and lettuce intercropping as a function of the time of lettuce transplantation. The indifference of cucumber to the presence of lettuce can be attributed to crop traits such as fast growth and canopy formation in a stratum higher than that of

lettuce, as cucumber cultivation was guided vertically. Species in intercropping systems benefit from having leaves in a higher stratum, where light intensity is higher, decreasing and/or nullifying the interference of the coexisting species [35–38]. These characteristics classify cucumber as the dominant, most aggressive crop in the competition between these two intercropped species [3,39].

LER differed for the two lettuce cultivars when intercropped with cucumber due to the different intensities of the effect of the cucumber on 'Vanda' and 'Lucy Brown'. All treatments of 'Vanda' intercropped with cucumber produced commercial-quality fruit, although the heads were smaller. The LERs for intercropping 'Vanda' with cucumber were between 1.55 and 1.71, with the lowest LER for the cucumber population density of 55% of 2.35 plants m$^{-2}$, increasing as the population density increased (Figure 7). The contribution of the 'Vanda' lettuce to LER was highest at the lowest density, due to the higher PAR relative to the other densities. The contribution of cucumber, though, was lowest at the lowest density (Figure 8), because fewer cucumbers were produced due to the fewer plants per area (Figure 6).

The LERs for the intercropping of 'Lucy Brown' lettuce with cucumber were between 1.30 and 1.49, with the lowest LER for the cucumber population density of 100% of 2.35 plants m$^{-2}$, which increased as the population density decreased (Figure 7). The commercialization of 'Lucy Brown' lettuce corresponds to the sale of loose or processed (chopped) leaves [40–42], despite the lower LERs than when intercropped with 'Vanda'. LERs >1 produced yields about 30 and 49% higher than for monocultures with the same area. Commercialization, however, requires head formation, so only intercropping at the lowest density was feasible. The contribution of 'Lucy Brown' to LER had the same pattern as that described for 'Vanda', but with a higher intensity than for 'Vanda' as cucumber population decreased. 'Vanda' increased its LER contribution by 40%, from 0.50 (100%) to 0.71 (55% of 2.35 cucumber plants m$^{-2}$), and 'Lucy Brown' increased its contribution by 183% (0.18 to 0.51).

The lettuce cultivars did not significantly affect the contribution of cucumber to LER (Table 5). The contributions of 'Vanda' and 'Lucy Brown' to the LER were <1 (Figure 8), corroborating the results of Cecílio Filho et al. [3,30] in studies evaluating the intercropping of lettuce with tomato and cucumber.

The contributions of cucumber to the LERs were inversely proportional to the LERs of the lettuce cultivars, as the yield per cucumber plant increased as the population density decreased. The increase in fruit yield per plant, however, could not compensate for the fewer plants per area, which decreased the yield. Fruit yield per area was used for calculating LER, disregarding fruit yield per plant.

## 5. Conclusions

The yields of the lettuce cultivars 'Vanda' and 'Lucy Brown' were negatively affected by intercropping and were affected more by higher cucumber population density. The cucumber yield per plant and per area were not influenced by the presence of lettuce. Population density directly influenced the amount of light that reached the lettuce plants and consequently their growth. Intercropping used land more efficiently than the monocultures of lettuce and cucumber, but more for 'Vanda' than 'Lucy Brown'.

**Author Contributions:** This work is a combined efforts of all the authors; Conceptualization, A.B.C.F.; Data curation, J.C.B., G.d.S.R.; Formal analysis, R.G.T.R., J.C.B., G.d.S.R.; Investigation, R.G.T.R., A.F.D.; Methodology, A.B.C.F., R.G.T.R., G.d.S.R.; Resources, A.B.C.F.; Supervision, A.B.C.F.; Writing - original draft, R.G.T.R., A.F.D.; Writing - review & editing, A.B.C.F. All authors have read and agreed to the published version of the manuscript.

**Funding:** This research received no external funding.

**Acknowledgments:** We thank the Coordination of Improvement of Higher Level Personnel (CAPES) for the scholarship granted to the first author to complete a doctorate degree course.

**Conflicts of Interest:** The authors declare no conflict of interest.

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
