# Peer review of "Land Equivalent Ratio in the Intercropping of Cucumber with Lettuce as a Function of Cucumber Population Density"

_agriculture, doi:10.3390/agriculture10030088_

Round 1

Reviewer 1 Report

Dear Authors,

The paper Is very-well written and arranged. The topic Is also interesting and novel. When I read It I was surprised about the good work.

However from my point of view the introduction is too small respect to the other paragraphs, so my suggestion is to balance that section by improving It with more references (Only two papers where at the moment, analysed in the state of the art). I guess that the topic and scope of research could be better introduced with a huge overview.

Author Response

March 12th, 2020

To

Dr. Audrey Wang

Editor

Dear Editor,

As requested in your correspondence, we are forwarding our replies to each observation raised by the Reviewers. The identification of the manuscript is Agriculture 711900.

We wish to thank the Reviewers, mainly for hard work of the Reviewer 2, for their pertinent notes and comments about our manuscript. We also want to thank you for believing that our manuscript has potential to be published in this prestigious journal, which would be a great honor for us.

In the manuscript named Agriculture 711900 – Tracked version, the Editor can find all changes that we have done to attend the Reviewers, and they can be easily understood according explanations to follow. The responses to the comments requested in the file are:

 TITLE:

Reviewers: no requests.

ABSTRACT

Reviewers: no requests.

KEYWORDS:

Reviewers: no requests.

INTRODUCTION

Reviewer 1 and 2: Basically, the two reviewers requested us to increase the item including more references. We understood the concerning of the reviewers, and we included information that have highlighted in the text.

Reviewer 2

Line 30: It is rather speculative to say that “higher yield per unit area is the greatest advantage of intercropping”, since there are several potential other advantages too and the main advantage might vary case-specifically. So I suggest to rephrase this sentence accordingly.

ANSWER: We excluded this sentence and replace it with: “Intercropping is a technology that enables production with rational land use and less environmental impact [1]. Besides of intercropping presents higher biological diversity and rapid soil cover [2] it allows greater efficiency in the use of agricultural inputs and labour, and helps increase the income from agricultural activity [3]. However intercropping is a more complex production system than monoculture [1] and its…”

Line 32: “differ in cycle” do you mean growth rhythm? “nutrients” do you mean in nutrient demand or uptake or content? Please specify.

ANSWER: No, cycle length and nutrient demand. We put “length” and “demand” in the text. Thank you for your observation.

Line 44: Please define LER better at first mention: what does it stand for and how is it calculated? Is it suitable and often used for greenhouse studies?

ANSWER: We couldn’t include an explanation about LER where the Reviewer requested. But in the Materials and Methods, item 2.4.5 we put the information there. In the same item, it is showed how LER is calculated. Yes, this index can be used to compare land use efficiency also for greenhouse because you have both systems (monoculture and intercropping) in that place.

Line 53: Please define on which crops (refs 11 & 12).

ANSWER: That’s right. We included at the end of that paragraph. (tomato and lucerne).

Line 54-55: Please add study hypothesis and main aim of the study.

ANSWER: We included hypothesis before the study objective. We understand that the objective of the study is clear. We believe that the inclusion of the hypotheses will improve the understanding of the objective.

Overall, some more background info on the use of growth resources and types of intercropping studies done in greenhouses could be added.

ANSWER: We couldn’t find research about cucumber-lettuce intercropping in greenhouse. So with all respect to Reviewer, we do not consider appropriated to use examples about other species intercropped to support our study. 

MATERIALS AND METHODS

Reviewer 2

Line 58-59 How long was the experiment? Do you think that there should have been multiple replications of the whole experiment in order to verify the findings? How representative is the experiment for greenhouse cultivation in general?

ANSWER: The information about the experimental period was completed and it has put at the end of first sentence of the item. Thank you for your observation. Really, it was absent the information about the end of experiment.

No, we understand that the results wouldn’t be modified if the experiment were replicated at the same time of the year because it is an issue about the cucumber effect on lettuce. Of course, if we decide to carried out another experiment in another season, we believe that the results can be different, but it is another study to compare effect of season (weather) on intercropping efficiency. Sorry, but we believe that the study was representative of a crop production in all its steps.

Line 86-90: Please specify the used fertilization more precisely, not only citing earlier work.

ANSWER: That´s right. The information requested has added to text starting in line 105 (Tracked version).

Line 247: Words missing from the first sentence: “LER interact…”? Please rephrase.

ANSWER: Thank you for observation. It was corrected (line 292 – Tracked version)

RESULTS

Reviewers: no requests.

DISCUSSION

Reviewer 2

could be more compact

ANSWER: We prefer to keep it like this.

CONCLUSION

Reviewer 2

Too much repetition of results, please rephrase to compress and conclude the main findings and recommendations.

ANSWER: We agree with the Reviewer. We eliminated some sentences, which were simple results and we maintain only those that answer to the hypotheses

REFERENCES

According to request of both Reviewers to increase and rewrite some parts of the Introduction item, we did it. So we include more two references and the first reference of the first version was eliminated. Due it was necessary renumber all citations in the text and references item.

References (number)

Old version

New version

1

excluded

1 (included)

2 (included)

5

3 (moved)

3

4

4

5

6

6

7

7

8…29

8…29

2

30

30

31

31

32

32

33

33 … 42

34..43

FIGURES:

Reviewers: no requests.

ANSWER: Besides the error hasn’t been observed by Reviewers, we replaced the Figure 1 and 7. The figures content were not modified. There were changes in the X and Y axis title, respectively.

TABLES
Reviewers: no requests.

We hope we have answered all the requests. We are available in case of new corrections or if necessary any more explanation. Due both Reviewers have recognized that English language is good, we didn´t send to a language specialist. But if Editor understand necessary we can send to be reviewed by MDPI services.

Arthur Bernardes Cecílio Filho

Reviewer 2 Report

The manuscript describes a greenhouse experiment on intercropping lettuce with cucumber. Overall, using intercropping in greenhouses is novel and interesting, and the study provides valuable information on this. The background could involve more references from earlier work in greenhouses and discuss how they compare to field conditions. The results are clearly presented. Discussion and Conclusions could benefit from compressing and emphasizing the main findings. I include some more specific suggestions for revision below:

Introduction

  1. L. 30: It is rather speculative to say that “higher yield per unit area is the greatest advantage of intercropping”, since there are several potential other advantages too and the main advantage might vary case-specifically. So I suggest to rephrase this sentence accordingly.
  2. L. 32: “differ in cycle” do you mean growth rhythm? “nutrients” do you mean in nutrient demand or uptake or content? Please specify.
  3. L. 44: Please define LER better at first mention: what does it stand for and how is it calculated? Is it suitable and often used for greenhouse studies?
  4. L. 53: Please define on which crops (refs 11 & 12).
  5. L. 54-55: Please add study hypothesis and main aim of the study.

Overall, some more background info on the use of growth resources and types of intercropping studies done in greenhouses could be added.

Materials and methods

  1. L. 58-59 How long was the experiment? Do you think that there should have been multiple replications of the whole experiment in order to verify the findings? How representative is the experiment for greenhouse cultivation in general?
  2. L. 86-90: Please specify the used fertilization more precisely, not only citing earlier work.
  3. L. 247: Words missing from the first sentence: “LER interact…”? Please rephrase.

Discussion could be more compact: it includes now quite many references to the Results section, but it could be compressed to more clearly emphasize the main findings and their relation to earlier work.

Conclusions: Too much repetition of results, please rephrase to compress and conclude the main findings and recommendations.

References: Please remove the double numbering in the reference list.

Author Response

(The authors gave the same response as above.)
